# Ensemble Learning of Multiple Models Using Deep Learning for Multiclass Classification of Ultrasound Images of Hepatic Masses

**DOI:** 10.3390/bioengineering10010069

**Published:** 2023-01-05

**Authors:** Norio Nakata, Tsuyoshi Siina

**Affiliations:** 1Division of Artificial Intelligence in Medicine, Center for Integrated Medical Research, The Jikei University, School of Medicine, 3-25-8 Nishi-Shinbashi, Minato-ku, Tokyo 105-8461, Japan; 2Shibaura Institute of Technology, Graduate School of Science and Engineering, 3-7-5 Toyosu Koto-ku, Tokyo 135-8548, Japan

**Keywords:** ensemble learning, deep learning, convolutional neural network, liver, ultrasonography, artificial intelligence

## Abstract

Ultrasound (US) is often used to diagnose liver masses. Ensemble learning has recently been commonly used for image classification, but its detailed methods are not fully optimized. The purpose of this study is to investigate the usefulness and comparison of some ensemble learning and ensemble pruning techniques using multiple convolutional neural network (CNN) trained models for image classification of liver masses in US images. Dataset of the US images were classified into four categories: benign liver tumor (BLT) 6320 images, liver cyst (LCY) 2320 images, metastatic liver cancer (MLC) 9720 images, primary liver cancer (PLC) 7840 images. In this study, 250 test images were randomly selected for each class, for a total of 1000 images, and the remaining images were used as the training. 16 different CNNs were used for training and testing ultrasound images. The ensemble learning used soft voting (SV), weighted average voting (WAV), weighted hard voting (WHV) and stacking (ST). All four types of ensemble learning (SV, ST, WAV, and WHV) showed higher values of accuracy than the single CNN. All four types also showed significantly higher deep learning (DL) performance than ResNeXt101 alone. For image classification of liver masses using US images, ensemble learning improved the performance of DL over a single CNN.

## 1. Introduction

Hepatocellular carcinoma (HCC) is the most common primary liver cancer, the sixth most common cancer worldwide, and the second leading cause of cancer death, resulting in almost 800,000 annual deaths worldwide [1,2,3]. Surviving HCC is mainly influenced by the stages of disease at the time of diagnosis [4]. Ultrasonography (USG) is a simple, noninvasive tool with no risk of radiation exposure. USG is a safe, portable, relatively inexpensive, and easily accessible imaging modality, making it a useful diagnostic and monitoring tool in medicine [5,6]. Conventional USG is usually used as the first choice for the surveillance of HCC [4,7,8]. The reported sensitivity of US alone for HCC diagnosis ranges from 60 to 90%, with excellent specificity of over 90% [9,10]. Ultrasonography is often used to screen the liver, especially for HCC, due to its low cost, accessibility, and lack of X-ray exposure [11].

There are published studies of artificial intelligence (AI) in USG, including thyroid, breast, abdomen, pelvic area, obstetrics and gynecology such as fetus, heart and vascular. AI-based image classification of liver masses has been studied in computed tomography (CT) [12,13] and also AI-based image classification of colorectal cancer liver metastases has been studied in magnetic resonance imaging (MRI) [14]. Focal liver lesions (FLLs) are described as abnormal portions of the liver that are primarily derived from hepatocytes, biliary epithelium, and mesenchymal tissue [15]. Ultrasonography is the preferred method for the diagnosis of FLLs because it is inexpensive, noninvasive, and provides real-time imaging. This trend suggests that AI models using ultrasound images have advantages over CT and MRI in routine clinical applications [16]. For this reason, a relatively large number of research papers have been published on the application of ultrasound-based AI to FLLs, distinguishing benign from malignant [17,18,19,20,21,22], HCC, metastatic liver tumors [23], or in addition to these, hemangiomas, focal nodular hyperplasia (FNH), non-neoplastic lesions, focal fatty infiltration, cyst, etc. [24,25,26,27,28,29,30,31,32,33,34] These papers mainly attempt image classification using CNN [17,18,25,26,34], other neural networks [24,28], support vector machine (SVM) [19,20,23], and radiomics [29,30,31,32,33]. There were no papers on the application of AI with ultrasound to FLLs using ensemble learning.

Ensemble learning, which uses multiple learning models, has recently become popular in image classification using DL. The purpose of using ensemble learning is to improve the AI performance by having multiple models cooperate with each other. Conventional ensemble learning related to diagnostic imaging includes studies using voting and stacking methods [35,36,37,38,39,40,41,42,43,44,45,46,47,48]. Ensemble pruning methods, known as ensemble selection methods, aim at reducing the complexity of ensemble models. One of the typical pruning methods is the ranking method. The ranking method rank the individual members based on a predetermined criterion and select the top ranked base models according to a threshold [49]. There are studies that performed ensemble learning by selecting the top 3 models out of 9 trained CNN models [35], and a study using 15 trained CNNs and ensemble learning from all of them [37]. Both of these two papers use the ranking method, but neither provides a rationale as to why the optimal rank was selected, and there is no discussion of ensemble pruning. In a study of deep ensemble learning for coronavirus disease 2019 (COVID-19) detection, the results of all seven learning models confirmed that the performance of the top three ensembles was superior to that of the top five and top seven ensembles [47].

The purposes of this study are as follows. (1) To compare the usefulness of several ensemble learning methods using multiple CNN learning models for multiclass image classification of FFLs in US images. (2) To investigate the usefulness of one of the ensemble pruning methods, the ranking method, for multiclass image classification of FFLs in US images.

## 2. Materials and Methods

### 2.1. Dataset

This is a multicenter diagnostic study using the B-mode ultrasound image dataset of FLLs. The inclusion criteria for data collection are as follows: (1) Patients’ FLLs were detected by B-mode US. (2) The diagnosis of the FLLs was confirmed via histology, contrast-enhanced computed tomography (CECT), contrast-enhanced ultrasonography (CEUSG) using Sonazoid, or gadoliniumethoxybenzyl-diethylenetriamine pentaacetic acid (Gd-EOB-DTPA)-enhanced MRI. In this study, 101,242 B-mode images of FFLs from 30,873 cases were collected from 1 April 2018 to 25 July 2021 in 11 core medical centers through a research project funded by the Japan Agency for Medical Research and Development (AMED), entitled “National Database Construction of Digital Ultrasound Images and Artificial Intelligence-Assisted Ultrasound Diagnostic System Development” [16]. Of these images, this study included 26,440 ultrasound B-mode, grayscale still images of 6056 cases of FLL that were selected. The images used in this project were classified into four categories: benign liver tumor (BLT) 6320 images from 1564 cases, liver cyst (LCY) 2560 images from 1018 cases, metastatic liver cancer (MLC) 9720 images from 1469 cases, primary liver cancer (PLC) 7840 images from 2005 cases. After randomly sorting test images by each class, 250 test images were again randomly selected for each class, for a total of 1000 test images (Figure 1), and the remaining images were used as training images. The 250 test images in each of the four classes were collected from 227 BLT cases, 200 LCY cases, 210 MLC cases, and 191 PLC cases. In order to check for similarity in test images of lesions collected from the same case, the test images for each class were hashed and the Hamming distance was measured to ensure that there were no similar images. Similarity was judged as not similar if the bit agreement between the hash values of the two images was less than 80% [50]. The training images consisted of 6970 BLT images, 2310 LCY images, 9470 MLC images, and 7590 PLC images. Pixel information in US grayscale images is provided by numbers ranging from 0 to 255, with 0 being white and the numbers becoming black as they approach 255. All US image data used were standardized by arraying, converting the data type to a 64-bit floating-point number, and then dividing by 255, with the pixel information being a number in the range 0 to 1.The study was approved by the Institutional Ethical Committee.

### 2.2. Hardware and Software

The hardware used was a desktop computer created for AI. The specifications were Intel core i9 CPU, 128 GB random access memory(RAM), and NVIDIA RTX A6000 graphics processing unit (GPU). Ubuntu Linux version 20.04 (https://jp.ubuntu.com/ accessed on 20 November 2022) was used as the software. Programming languages used were python version 3.8 (https://www.python.org/ accessed on 20 November 2022) and tensorflow version 2.8 (https://www.tensorflow.org/ accessed on 20 November 2022), keras version 2.8 (https://keras.io/ accessed on 20 November 2022), a machine learning library, and scikit-learn 0.24 (https://scikit-learn.org accessed on 20 November 2022) were used to create each program.

### 2.3. Training of 16 CNNs

16 different CNN models were fine-tuned by replacing only fully-connected layer, or by replacing part of the convolutional layer and fully-connected layer were used for training and testing ultrasound images (Figure 2). EfficientNetB0-B6 [51] were pre-trained on ImageNet with Noisy Student Training and were fine-tuned, and the other 9 models were pre-trained on ImageNet and were fine-tuned. Noisy Student Training extends the idea of self-training and distillation with the use of equal-or-larger student models and noise added to the student during learning. ImageNet first trains an EfficientNet model on the labeled images, which are used as a teacher to generate pseudo-labels for the 300 million unlabeled images. Next, a larger EfficientNet is trained as a student model for the combination of labeled and pseudo-labeled images. This process is iterated by returning the student as the teacher. During student learning, inject noise into the student, such as dropouts, probabilistic depth, and data augmentation with RandAugment, to ensure that the student generalizes better than the teacher [52]. 16 types of training were performed using the different models and the aforementioned training images. The resolution of all input images was set to 256 × 256 in order to compare them and ensure reproducibility of training results. The batch sizes were all set to 32, the number of epochs to 50, and the random number seed was fixed to an arbitrary seed value. Width shift, height shift and horizontal flip were used as data augmentation. The floating-point number for width shift and height shift was set to 0.1. K-partition cross-validation (k = 10) was used as a method to validate the predictive performance of the machine learning models (Figure 2). The model fitting callback was the early stopping function of Keras, which stops training before overtraining occurs, and training was stopped if there was no improvement during 10 epochs of val_loss values. We also reduced the learning rate by 0.1 if there was no improvement for 3 epochs using ReduceLROnPlateau in Keras.

### 2.4. Classification Metrics of Test Results for 16 CNNs

After training with the training set images, 16 CNN models were tested with 1000 test set images and probabilities of four classes and predicted classes of all 1000 images were output (Figure 2). Performance metrics for multiclass classification are first calculated using the confusion matrix to calculate true positive (TP), true negative (TN), false positive (FP), and false negative (FN), and then these are used to calculate values for precision, sensitivity (recall), specificity, and f1 score. The calculation of TP, TN, FP, and FN in a multiclass is described below. TP: The true positive value is where the actual value and predicted value are the same; FN: The false-negative value for a class will be the sum of values of corresponding rows except for the TP value; FP: The false-positive value for a class will be the sum of values of the corresponding column except for the TP value; TN: The true negative value for a class will be the sum of values of all columns and rows except the values of that class that we are calculating the values for.

Using Table 1 as a calculation example, the TP, TN, FP, FN for the class BLT in multi-class are calculated as follows:

TP: The actual value and predicted value should be the same. Concerning BLT class, the value of cell 1 is the TP value.
TP_BLT_ = cell1

FN: The sum of values of corresponding rows except the TP value.
FN_BLT_ = (cell2 + cell3 + cell4)

FP: The sum of values of corresponding column except the TP value.
FP_BLT_ = (cell5 + cell9 + cell13)

TN: The sum of values of all columns and row except the values of that class that we are calculating the values for.
TN_BLT_ = (cell6 + cell7 + cell8 + cell9 + cell10 + cell11 + cell12 + cell13 + cell14 + cell15 + cell16)

TP, TN, FP, and FN are calculated in the same way for the remaining three classes.

In the case of multiclass classification, one class is selected as positive and the other classes are selected as negative, and the evaluation values are calculated. There are two types of averages: micro-average and macro-average. When the number of samples in each class differs greatly, the macro-average may not be able to calculate the actual accuracy, and the micro average may be selected. In this study, the number of samples for each class is the same, so the macro-average is used.

The precision, sensitivity, specificity, f1 score and accuracy for the class BLT are given by the following equations [53,54].
$${\mathrm{Precision}}_{\mathrm{BLT}}=\frac{{\mathrm{TP}}_{\mathrm{BLT}}}{{\mathrm{TP}}_{\mathrm{BLT}}+{\mathrm{FP}}_{\mathrm{BLT}}}$$
$${\mathrm{Sensitivity}}_{\mathrm{BLT}}=\frac{{\mathrm{TP}}_{\mathrm{BLT}}}{{\mathrm{TP}}_{\mathrm{BLT}}+{\mathrm{FN}}_{\mathrm{BLT}}}$$
$${\mathrm{Specificity}}_{\mathrm{BLT}}=\frac{{\mathrm{TP}}_{\mathrm{BLT}}}{{\mathrm{FP}}_{\mathrm{BLT}}+{\mathrm{TN}}_{\mathrm{BLT}}}$$
$$F1{\;\mathrm{score}}_{\mathrm{BLT}}=\frac{2\times {\mathrm{Sensitivity}}_{\mathrm{BLT}}\times {\mathrm{Precision}}_{\mathrm{BLT}}}{{\mathrm{Sensitivity}}_{\mathrm{BLT}}+{\mathrm{Precision}}_{\mathrm{BLT}}}$$
$${\mathrm{Accuracy}}_{\mathrm{BLT}}=\frac{{\mathrm{TP}}_{\mathrm{BLT}}+{\mathrm{TN}}_{\mathrm{BLT}}}{{\mathrm{TP}}_{\mathrm{BLT}}+{\mathrm{TN}}_{\mathrm{BLT}}+{\mathrm{FP}}_{\mathrm{BLT}}+{\mathrm{FN}}_{\mathrm{BLT}}}$$

These values can be calculated for each class. The macro-average is the average of these values. The formula for calculating the macro-average of precision (Precision_Macro_) in this study is as follows.
$${\mathrm{Precision}}_{\mathrm{Macro}}=\frac{{\mathrm{Precision}}_{\mathrm{BLT}}+{\mathrm{Precision}}_{\mathrm{LCY}}+{\mathrm{Precision}}_{\mathrm{MLC}}+{\mathrm{Precision}}_{\mathrm{PLC}}}{4}$$

Macro averages for other evaluation metrics are also calculated using the same formula as above. The 95% confidence intervals for precision, sensitivity, specificity, and accuracy were all calculated using the Newcombe method [55].

The receiver operating characteristic (ROC) curve and ROC area under the curve (AUC) score are not immediately applicable to a multiclass classifier. One vs. Rest, where each class is compared to the other classes simultaneously, allows drawing ROC curves for multiclass classifiers. 16 CNNs with 4 classes had ROC curves AUCs computed. Macro averages of the AUCs were then also computed for each model [56,57]. Finally, for each of the 16 CNN models, the macro-averaged accuracy values of the test results were ranked and sorted from highest to lowest.

### 2.5. Correlation Matrix for Each of the 16 CNN Test Results

A quantitative measure of the ensemble’s effectiveness is to look at the Pearson’s correlation coefficient (CORR) between classifiers. The CORRs of the estimated labels of 16 CNN classifiers are calculated.

### 2.6. Ensemble Learning

#### 2.6.1. Four Types of CNN Ensemble Learning

Since most of the previous papers [35,36,37,38,39,40,41,42,43,44,45,46,47,48] on ensemble learning with CNNs for medical images used either simple soft voting (SV), weighted average voting (WAV), weighted hard voting (WHV), or stacking(ST), we used four types of ensemble learning in this study: SV, WAV, WHV, and ST. Ensemble learning includes voting, bagging, boosting, and ST [58,59,60]. Voting includes SV and simple hard voting (majority voting).

##### Voting Ensemble including WHV, SV, and WAV

Hard voting is a simple majority voting image classification in which each classifier has one vote. In the case of hard voting, in order to obtain the optimal number of ensemble pruning based on the ranking method described in the next section, using an even number of classifiers may result in a tie number of votes for the four classes, and the same number of votes may make the class determination for image classification impossible. To avoid this situation and improve the performance of ensemble learning, there is a technique called WHV, also known as Weighted Majority Voting, which assigns different weights to each classifier. WHV, weighted majority voting is computed by associating a weight $w_{j}$ with classifier $C_{j}$:$$\hat{y}=arg\underset{i}{max}\overset{m}{\underset{j=1}{\sum }}w_{j}\chi_{A}\left(C_{j}\left(x\right)=i\right),$$
where $\chi_{A}$ is the characteristic function $C_{j}\left(x\right)=i\;\in A$, and A is the set of unique class labels. WHV is a method that assigns different weights to each classifier [61]. SV also has a method called WAV, which aims to improve performance by optimizing each classifier with different weights [48]. In SV, we predict the class labels based on the predicted probabilities p for classifier.
$$\hat{y}=arg\underset{i}{max}\overset{m}{\underset{j=1}{\sum }}w_{j}p_{ij},$$
where $w_{j}$ is the weight that can be assigned to the *i* th classifier.

In simple unweighted SV, the weight $w_{j}$ is assigned to 1. WAV also assigns different weights to each classifier. WHV and WAV require optimization of the respective weights. The difference between WHV and WAV is illustrated in Figure 3, citing two examples. There are some methods for weight optimization, such as grid search, random search, and Bayesian optimization [62,63]. In this study, we optimized the weighting for both WHV and WAV using Bayesian optimization with our own Python code.

##### Stacking Ensemble

In ST, the 16 CNN models are first trained together as a basic learner to obtain predictions. Then, the individual predictions are treated as the next set of training data and added from another layer called the meta-learner (Figure 4). There are also studies that use extreme gradient boosting (XGBoost), a representative method of gradient boosting, as a meta-learner for ST [64,65]. In this study, we employed the ST method using light gradient boosting machine (LightGBM) version 3.2.1 (https://lightgbm.readthedocs.io/ accessed on 20 November 2022.) as the meta-learner, which is a newer method of gradient boosting than XGBoost and is expected to have higher performance [66,67] (Figure 4). For LightGBM, k = 5 was set using k-partition cross-validation. The number of gradient boosting iterations (num_boost_round) was set to 1000, and the number of times that training was terminated (early_stopping) was set to 50 if the score did not improve a certain number of consecutive times in the evaluation data, even if the specified training count was not reached. The multiclass log loss (multi_logloss) was used as the metric for the LightGBM evaluation function. Other parameters of LightGBM were optimized by Optuna version 2.10.0 (https://www.preferred.jp/ja/projects/optuna/ accessed on 20 November 2022), an open-source software framework for automating the optimization of hyperparameters, which is a type of Bayesian optimization. We also fixed the random number seeds in the program to ensure reproducibility when using Optuna with LightGBM.

#### 2.6.2. Ensemble Pruning and Evaluation Metrics

As in previous papers on ensemble learning using CNNs for medical images [35,37,47], we employed a ranking method, a type of ensemble pruning, in which the top 1 classifier is selected from the classifiers ordered by the highest accuracy value obtained from the test results of each of the 16 CNNs alone. In this study, in order to find the optimal number of selected classifiers for pruning, we first started with ensemble learning using all 16 types of classifiers. Next, pruning was performed by decreasing the number of classifier members one by one, starting with the top 16 with the lowest accuracy. Finally, ensemble learning was performed using only top1 and top2, for a total of 15 types of ensemble learning. By running each of ensemble pruning for SV, WAV, WHV, and ST, 60 ensemble learning models were created (Figure 5). As with the testing of the 16 CNNs, 60 patterns of ensemble learning models were tested using 1000 B-mode FFLs US images of the test set. TP, TN, FP, FN, precision, sensitivity, specificity, f1 score, and accuracy were calculated as the evaluation metrics. since WHV is a kind of majority voting and outputs only prediction classes, it is usually not possible to draw ROC curves. Therefore, ROC-AUC values were calculated with SV, WAV, and ST.

### 2.7. Statistical Examination

McNemar’s test is used when there is a need to compare the performance of two classifiers [68,69]. The McNemar’s test is designed to focus primarily on the differences between the two classifiers, i.e., cases predicted in different ways. An example of the McNemar’s test for comparing two classifiers (classifier 1, classifier 2) is shown below. First, the correct answer classes of the test set and the predicted classes of each of classifier1 and classifier 2 are tabulated. These are compared to each other to calculate the number of cases correctly classified by both classifiers, the number correctly classified by classifier 1 but not by classifier 2, the number correctly classified by classifier 2 but not by classifier 1, and the number misclassified by both classifiers, respectively. These are used to create a contingency table (Table 2). Finally, since the McNemar’s test is essentially a form of Chi-square test with correspondence, we calculate the Chi-square values using the following formula.
$$\chi^{2}\;=\;\frac{{\left(n_{01}-n_{10}\right)}^{2}}{n_{01+}n_{10}}$$

The null hypothesis is that two cases do not agree by the same amount. If the null hypothesis is rejected, it suggests that the cases do not match in different ways and that there is evidence to suggest that the disagreement is skewed [67,68]. Given the selection of a significance level, the *p*-value calculated by the test can be interpreted as follows:

alpha: significance level, H0: null hypothesis.*p* > alpha: fail to reject H0, no difference between classifier 1 and classifier 2.*p* ≤ alpha: reject H0, significant difference between classifier 1 and classifier 2.

Based on this method, the following statistical analyses were conducted.

#### 2.7.1. Comparison of 16 Different CNNs

The McNemar test of statistical significance between the Top 1 model with the highest accuracy value and the results of each of the other best 2 to 16 CNN models among the 16 single CNN models.

#### 2.7.2. Comparison of Stand-Alone CNN and Ensemble Models

The method with the highest accuracy value among the SV, ST, WAV, and WHV ensemble learning models was selected. The McNemar test was then performed between the test results of all 15 members of the ensemble pruning from top2 to top16 of the best selected ensemble learning method and the test results of the Top1 CNN model with the highest accuracy value among the 16 CNNs.

## 3. Results

### 3.1. Evaluation of Test Results for 16 CNN Models

Four-class confusion matrix in 16CNNs are shown in Figure 6. True positive, false positive, false negative, and true negative values for 4-class classification in 16 CNNs are shown in Table 3. The evaluation metrics for each of the 16 CNNs are shown in Table 4 and Table 5. Among the 16 CNNs, ResNext101 (0.719) had the highest accuracy. With the exception of EfficientNetB0 (0.694), which had the 6th highest accuracy, the CNNs with the 4th through 9th highest accuracy were ResNet-based. The CNNs from the 11th to the lowest accuracy were EfficientNet-based. The CNN with the highest f1 score was ResNeXt101, consistent with the accuracy ranking. When comparing the values of precision, sensitivity, and specificity, all 16 ranking CNNs were high in the order of specificity, precision, and sensitivity. Comparing the test results of ResNeXt101, which ranked first in accuracy, with the other 15 CNN models, there was no significant difference in test result performance between the second to ninth place rankings. On the other hand, there was a significant difference (*p* < 0.05) between ResNeXt101 and each of the CNNs ranked 10th through 16th (Table 5). The CORRs among the estimated labels of the 16 CNN classifiers totaled 120. Of these, 8 were in the 0.3 range and 112 were 0.4 or higher. All 120 CORRs were below 0.95 (Table 6).

### 3.2. Evaluation of Test Results for Ensemble Learning and Ensemble Pruning

A comparison of the accuracy values for the ensemble learning is shown in Figure 6. For SV, top16 was used for SV (SV16) with the highest accuracy of 0.776. For ST, top9 (ST9) had the highest accuracy at 0.776. Referring to Table 5, this ST9 refers to ST using a total of nine different classifiers: ResNeXt101, Xception, InceptionResNetV2, SeResNeXt50, ResNeXt50, EfficientNetB0, SeResNeXt101, ResNet101, and ResNet50, starting from top1. For WHV, top13 (WHV13) had the highest accuracy at 0.779. top7 (WAV7) had the highest accuracy for WAV at 0.783, which was the highest accuracy value for all methods. For WHV, top13 (WHV13) had the highest accuracy at 0.779.

Four-class confusion matrix of the models with the highest accuracy by pruning in each of the four types of ensemble learning are shown in Figure 7. TP, FP, FN and RN values for 4-class classification in the models with the highest accuracy by pruning in each of the four types of ensemble learning are shown in Table 7. The respective metrics for SV16, ST9, WAV7, and WHV13 are shown in Table 8. As well as accuracy, precision (0.790), sensitivity (0.783). and specificity (0.928) for WAV7, specificity (0.928), f1-score (0.786), and the macro-AUC (0.935) for ROC was the highest value for all methods calculated in this study.

Referring to Figure 6 and Table 8, WAV was chosen as the method with the best performance in the test results, among the four ensemble methods. Table 9 showed the statistical analysis of the test results for all WAV members from WAV2 to WAV16 and for Res-NeXt101, the top1 accuracy ranking for each CNN. For WAV6 to WAV16. the test results were significantly better than for ResNeXt101, a single CNN (*p* < 0.01).

The average computation time required to output the results from the test data of each of the 15 ensemble learning methods from top2 to top16 was 4.26 s for Soft Voting, 18 min 7.38 s for Stacking, 13 min 22.23 s for WAV, and WHV was 40 min 55.73 s.

### 3.3. Details of Image Classification of Ultrasound Images of Liver Masses

Table 10 showed the image classification results of 4 different liver masses for ResNeXt101, the model with the highest accuracy among 16 CNNs, and WAV7, the model with the highest accuracy among ensemble learning. All the indices (precision, sensitivity, specificity, f1-score, ROC-AUC) for both ResNeXt101 and WAV7 were highest for LCY, followed by BLT and PLC and lowest for MLC was the lowest. In comparison of the values of precision, sensitivity, and specificity, only the liver cyst of WAV7 had 100% precision and specificity, while all the others, both ResNeXt101 and WAV7, had 100% precision, specificity, specificity, sensitivity, and specificity. All others were high in the order of specificity, precision, and sensitivity for both ResNeXt101 and WAV7.

## 4. Discussion

In this study, we used the value of accuracy as a metric that is often used to compare multiple models to rank the test results of 16 different CNN models [54], When comparing the accuracy of multiple training models, the random seed needs to be fixed to ensure reproducibility of the test results. For this reason, we took this problem into consideration by fixing the random seed at multiple locations in the program as appropriate. However, since the main purpose of this paper is to focus on the comparison of multiple models and the usefulness of ensemble learning, fixing the random seed does not necessarily mean that the test results of the training model will output the best maximum value. It should be taken into account that each of the evaluation metrics listed tends to be somewhat lower. EfficientNet is newer than the resnet and inception-based CNN models, and EfficientNet is generally expected to perform better in ImageNet with Noisy Student training than ImageNet without Noisy Student [52]. For this reason, we employed ImageNet with noisy student training for pre-training of EfficientNet. However, even with EfficientNet, no better accuracy was obtained compared to the resnet and inception-based CNN models, except for EfficientNetB0. In the future, it seems necessary to examine the usefulness of EfficientNet by increasing or decreasing the number of images.

The type of model chosen to achieve high accuracy may also depend on the type and resolution of the target medical images. Therefore, a simple comparison of other papers on the usefulness of ensemble learning and on subjects that differ from each other is limited. In this study, 16 types were widely selected from inception, resnet, and EfficientNet systems. A quantitative measure of the effectiveness of ensemble is to look at the Pearson’s CORR between classifiers, and it is said that ensemble learning is expected to be effective when Pearson’s CORR is 0.95 or less [70]. The 16 CNNs used in this study all had CORRs of 0.95 or less, suggesting that ensemble learning can help improve prediction performance.

There was no significant difference between RexNeXt101, the CNN with the highest accuracy values, and the CNNs ranked 2nd through 9th in accuracy values. Therefore, ResNeXt101 was adopted as the representative of the single CNN for comparison of test results with ensemble learning. In the comparison of SV, ST, WAV, and WHV, WAV showed the best accuracy. As for ST, even though the gradient boosting method was used as a meta-model, it did not perform as well as SV and WAV, as in the previous paper on ensemble learning using medical images [36].

Ensemble pruning was also performed, and test results were compared between WAV, the best performing of the four ensemble learning methods, and ResNexT101, a single CNN. The results showed that WAV significantly outperformed ResNeXt101 up to WAV6, suggesting that pruning can reduce the number of classifiers that are members of the ensemble learning. However, further research is needed on pruning methods and parameters.

In this study, the average computation time with SV was 4.26 s, while WAV took 13 min. Advances in hardware have made faster computation speeds possible, making shorter processing times possible. The results of this study also show that ensemble learning clearly improves accuracy compared to single learning models, and it is expected that ensemble learning will become more popular in the future, as the trend toward improved computation speed is expected to continue.

Although very rare, cases of PLC associated with BLT have been reported, and US images have been included in these case reports [71,72]. It is believed that the general DL, including this study, cannot correctly diagnose such rare cases, which is a limitation of this study. Further studies are needed for computer assisted diagnosis of such rare diseases. However, if the DL model predicts BLT on one scan and PLC on another scan for the same lesion, it may provide a hint for radiologists to consider the possibility of rare cases for the paradoxical prediction of DL.

## 5. Conclusions

In multiclass classification of FLLs using US B-mode images, ensemble learning was shown to improve accuracy and the deep learning performance over a single CNN. It was thought that future research should explore ensemble pruning for medical images such as X-ray, CT and MRI, other than US images, especially to find out how many classifier members are useful in ensemble learning by ranking methods.

## Figures and Tables

**Figure 1 bioengineering-10-00069-f001:**
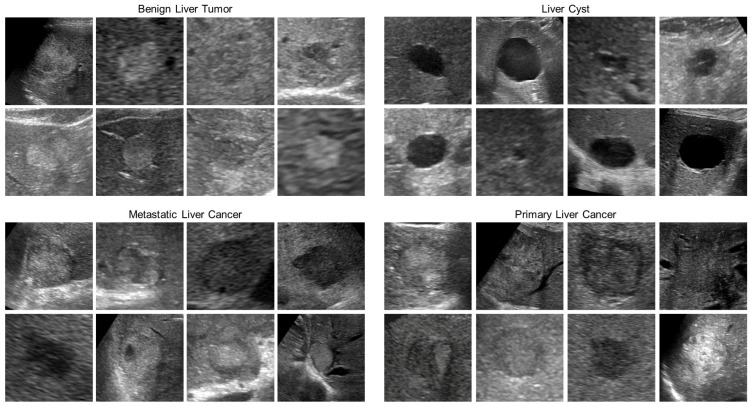
Example of test images. Each B-mode ultrasound image is a grayscale image of 256 × 256 pixels.

**Figure 2 bioengineering-10-00069-f002:**
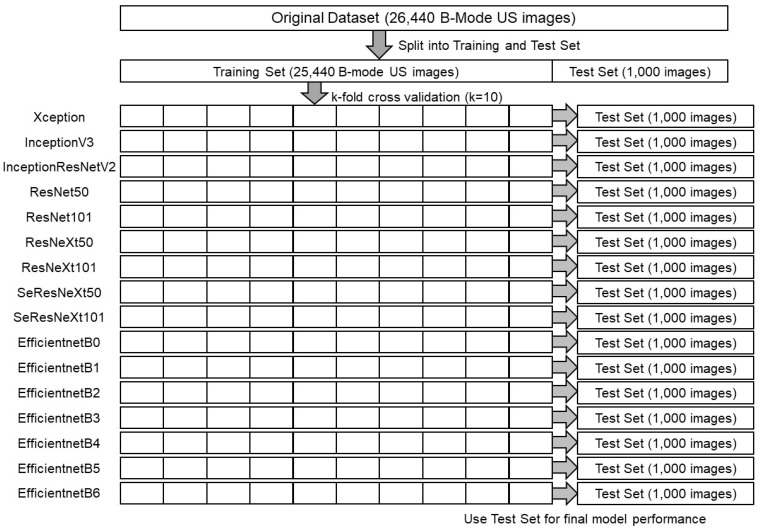
A schematic of training and tests of 16 CNNs.

**Figure 3 bioengineering-10-00069-f003:**
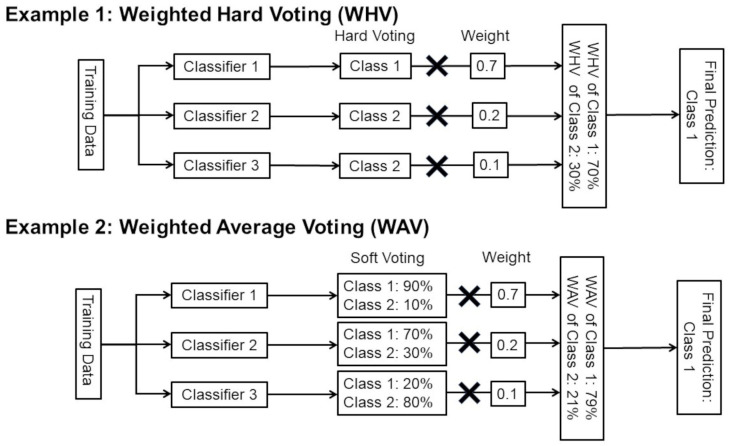
Examples of WHV and WAV. This figure shows examples of WHV and WAV with three different classifiers and two class classifications. In this study, 16 different classifiers and 4 class classifications were used.

**Figure 4 bioengineering-10-00069-f004:**
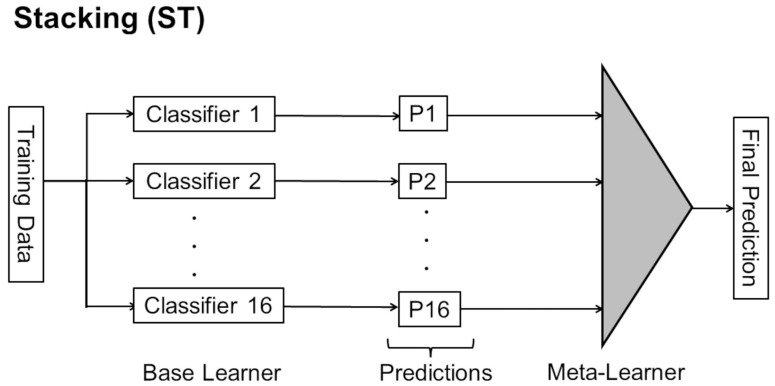
Description of ST. 16 different CNNs as base learner and LightGBM as meta-learner were used in this study.

**Figure 5 bioengineering-10-00069-f005:**
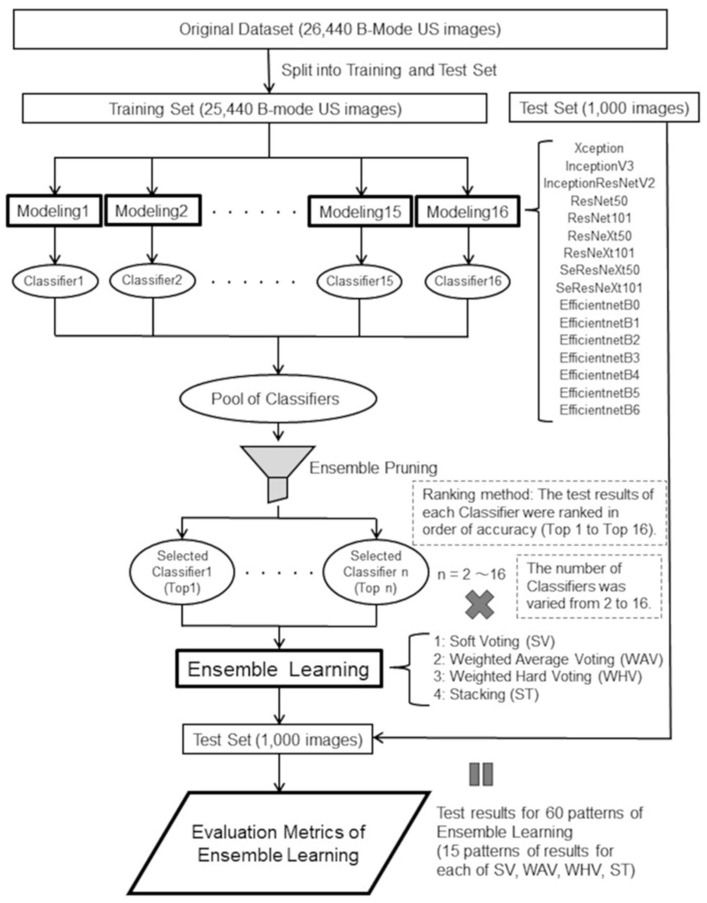
A schematic of ensemble pruning and evaluation metrics.

**Figure 6 bioengineering-10-00069-f006:**
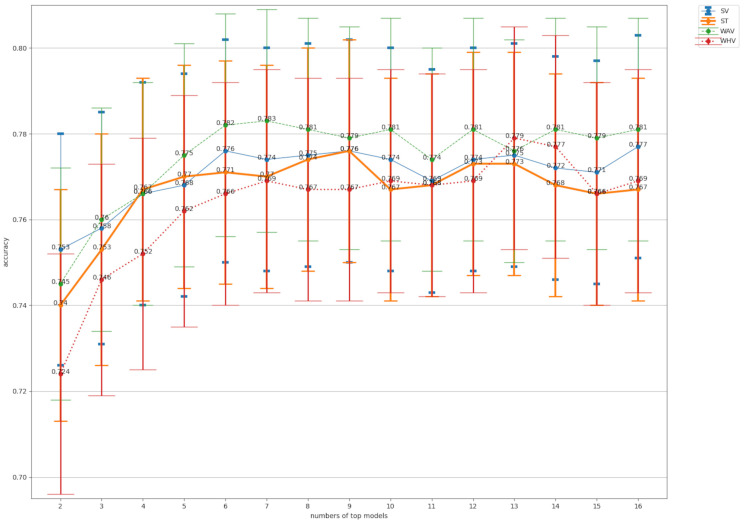
A comparison of the accuracy values for ensemble pruning. SV: soft voting, ST: stacking. WAV: weighted average voting, WHV: weighted hard voting. Numbers of top models represents the number of classifiers used for ensemble training from top 2 to top 16 in accuracy. The bars in the graph are 95% confidence intervals.

**Figure 7 bioengineering-10-00069-f007:**
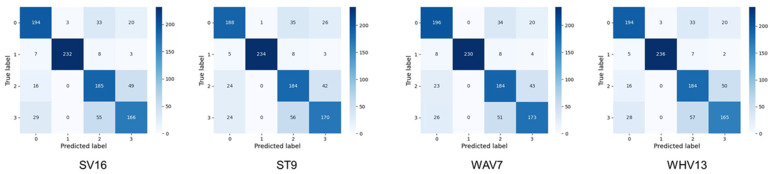
Four-class confusion matrix of the models with the highest accuracy by pruning in each of the four types of ensemble learning. The classes indicated by the numbers of the labels on the x- and y-axes of each confusion matrix are as follows. 0: Benign Liver Tumor, 1: Liver Cyst, 2: Metastatic Liver Cancer, 3: Primary Liver Cancer, SV16: top16 was used for soft voting, ST9: top9 was used for stacking, WAV7: top7 was used for weighted average voting, WHV13: top13 was used for weighted hard voting.

**Table 1 bioengineering-10-00069-t001:** Confusion matrix for multiclass classification. Actual values are the correct values of the ultrasound images of the test set. Prediction values are the predictions of the ultrasound images of the test set by the classifier. BLT: benign liver tumor, LCY: liver cyst, MLC: metastatic liver cancer, PLC: primary liver cancer.

		Predicted Values			
		BLT	LCY	MLC	PLC
Actual Values	BLT	cell1	cell2	cell3	cell4
	LCY	cell5	cell6	cell7	cell8
	MLC	cell9	cell10	cell11	cell12
	PLC	cell13	cell14	cell15	cell16

**Table 2 bioengineering-10-00069-t002:** Contingency table used for the McNemar’s test to compare two classifiers. n00: number of items classified correctly by both Classifier 1 and Classifier 2, n01: number of items classified correctly by Classifier 1 but not by Classifier 2, n10: number of items classified correctly by Classifier 2 but not by Classifier 1, n11: number of items misclassified by both Classifier 1 and Classifier 2. Null hypothesis: n01 = n10.

Contingency Table		
	Classifier 2 Correct	Classifier 2 Incorrect
Classifier 1 Correct	n00	n01
Classifier 1 Incorrect	n10	n11

**Table 3 bioengineering-10-00069-t003:** True positive, false positive, false negative, and true negative values for 4-class classification in 16CNNs. BLT: benign liver tumor, LCY: liver cyst, MLC: metastatic liver cancer, PLC: primary liver cancer, TP: true positive, FP: false positive, FN: false negative, TN: true negative.

		Xception	InceptionV3	InceptionResNetV2	ResNet50	ResNet101	ResNeXt50	ResNeXt101	SeResNeXt50	SeResNeXt101	EfficientNetB0	EfficientNetB1	EfficientNetB2	EfficientNetB3	EfficientNetB4	EfficientNetB5	EfficientNetB6
BLT	TP	180	156	169	170	167	173	172	194	190	165	153	179	151	134	148	156
	FP	58	52	67	74	70	62	59	116	105	61	100	114	75	84	64	79
	FN	70	94	81	80	83	77	78	56	60	85	97	71	99	116	102	94
	TN	692	698	683	676	680	688	691	634	645	689	650	636	675	666	686	671
LCY	TP	217	196	204	225	225	219	226	221	222	207	194	208	209	207	218	213
	FP	8	6	1	8	10	3	2	4	2	2	11	10	9	7	6	14
	FN	33	54	46	25	25	31	24	29	28	43	56	42	41	43	32	37
	TN	742	744	749	742	740	747	748	746	748	748	739	740	741	743	744	736
MLC	TP	166	169	164	184	183	156	166	160	146	161	144	124	165	126	151	159
	FP	129	123	123	162	159	132	127	110	103	114	142	98	162	145	129	147
	FN	84	81	86	66	67	94	84	90	104	89	106	126	85	124	99	91
	TN	621	627	627	588	591	618	623	640	647	636	608	652	588	605	621	603
PLC	TP	152	155	163	105	111	147	155	124	132	161	137	153	136	150	145	132
	FP	90	143	109	72	75	108	93	71	100	129	119	114	93	147	139	100
	FN	98	95	87	145	139	103	95	126	118	89	113	97	114	100	105	118
	TN	660	607	641	678	675	642	657	679	650	621	631	636	657	603	611	650

**Table 4 bioengineering-10-00069-t004:** The evaluation metrics for each of the 16 CNNs.

Model	Precision *^,^ **	Sensitivity *^,^ **	Specificity *^,^ **	Accuracy **	Ranking of Accuracy
ResNeXt101	0.731 (0.778, 0.686)	0.719 (0.772, 0.666)	0.906 (0.925, 0.888)	0.719 (0.747, 0.691)	1
Xception	0.728 (0.777, 0.679)	0.715 (0.769, 0.661)	0.905 (0.924, 0.886)	0.715 (0.743, 0.687)	2
InceptionResNetV2	0.720 (0.766, 0.675)	0.700 (0.756, 0.644)	0.900 (0.919, 0.881)	0.700 (0.728, 0.672)	3
SeResNeXt50	0.709 (0.758, 0.660)	0.699 (0.752, 0.646)	0.900 (0.919, 0.880)	0.699 (0.727, 0.671)	4
ResNeXt50	0.710 (0.758, 0.663)	0.695 (0.750, 0.640)	0.898 (0.917, 0.879)	0.695 (0.724, 0.666)	5
EfficientNetB0	0.715 (0.762, 0.669)	0.694 (0.750, 0.638)	0.898 (0.917, 0.879)	0.694 (0.723, 0.665)	6
SeResNeXt101	0.698 (0.746, 0.650)	0.690 (0.744, 0.636)	0.897 (0.916, 0.877)	0.690 (0.719, 0.661)	7
ResNet101	0.698 (0.750, 0.647)	0.686 (0.739, 0.633)	0.895 (0.915, 0.875)	0.686 (0.715, 0.657)	8
ResNet50	0.697 (0.748, 0.645)	0.684 (0.737, 0.631)	0.895 (0.914, 0.875)	0.684 (0.713, 0.655)	9
InceptionV3	0.705 (0.754, 0.656)	0.676 (0.733, 0.619)	0.892 (0.912, 0.872)	0.676 (0.705, 0.647)	10
EfficientNetB2	0.674 (0.725, 0.621)	0.664 (0.719, 0.607)	0.888 (0.909, 0.867)	0.664 (0.692, 0.634)	11
EfficientNetB5	0.680 (0.730, 0.630)	0.662 (0.718, 0.606)	0.887 (0.908, 0.867)	0.662 (0.691, 0.633)	12
EfficientNetB3	0.681 (0.733, 0.630)	0.661 (0.718, 0.604)	0.886 (0.908, 0.866)	0.661 (0.690, 0.632)	13
EfficientNetB6	0.673 (0.726, 0.620)	0.660 (0.716, 0.604)	0.887 (0.908, 0.866)	0.660 (0.689, 0.631)	14
EfficientNetB1	0.647 (0.700, 0.595)	0.628 (0.687, 0.569)	0.876 (0.898, 0.854)	0.628 (0.658, 0.598)	15
EfficientNetB4	0.638 (0.689, 0.588)	0.617 (0.675, 0.559)	0.872 (0.894, 0.851)	0.617 (0.647, 0.587)	16

* macro-average ** (high and low 95% confidence interval).

**Table 5 bioengineering-10-00069-t005:** The evaluation metrics for each of the 16 CNNs (Table 4 continued) and.significant difference test between ResNeXt101(top1) and 15 other models.

Model	F1 Score *	ROC Macro AUC	Ranking of Accuracy	*p*-Value **
ResNeXt101	0.724	0.902	1	–
Xception	0.720	0.900	2	0.882
InceptionResNetV2	0.707	0.880	3	0.377
SeResNeXt50	0.699	0.874	4	0.341
ResNeXt50	0.701	0.903	5	0.238
EfficientNetB0	0.701	0.866	6	0.224
SeResNeXt101	0.692	0.870	7	0.148
ResNet101	0.685	0.887	8	0.107
ResNet50	0.682	0.888	9	0.081
InceptionV3	0.684	0.859	10	0.037
EfficientNetB2	0.666	0.868	11	0.006
EfficientNetB5	0.668	0.890	12	0.005
EfficientNetB3	0.667	0.867	13	0.006
EfficientNetB6	0.664	0.877	14	0.004
EfficientNetB1	0.635	0.852	15	<0.001
EfficientNetB4	0.624	0.856	16	<0.001

* macro-average. **: *p*-value of the McNemar’s test to compare ResNeXt101(top1) with other 15 models.

**Table 6 bioengineering-10-00069-t006:** Correlation matrix for each of the 16 CNN test results.

ResNeXt101	1.000															
Xception	0.585	1.000														
InceptionResNetV2	0.536	0.584	1.000													
SeResNeXt50	0.599	0.519	0.462	1.000												
ResNeXt50	0.627	0.534	0.529	0.583	1.000											
EfficientNetB0	0.503	0.542	0.543	0.409	0.487	1.000										
SeResNeXt101	0.541	0.509	0.483	0.630	0.528	0.388	1.000									
ResNet101	0.558	0.530	0.505	0.569	0.556	0.442	0.532	1.000								
ResNet50	0.576	0.520	0.502	0.572	0.596	0.437	0.550	0.746	1.000							
InceptionV3	0.505	0.558	0.593	0.356	0.464	0.594	0.404	0.448	0.436	1.000						
EfficientnetB2	0.492	0.511	0.424	0.559	0.499	0.412	0.496	0.563	0.548	0.359	1.000					
EfficientnetB5	0.496	0.488	0.486	0.409	0.480	0.469	0.373	0.473	0.474	0.482	0.513	1.000				
EfficientnetB3	0.539	0.518	0.523	0.515	0.503	0.493	0.471	0.477	0.519	0.435	0.574	0.568	1.000			
EfficinetnetB6	0.524	0.515	0.429	0.462	0.480	0.427	0.420	0.512	0.491	0.450	0.548	0.613	0.506	1.000		
EfficientnetB1	0.447	0.484	0.424	0.434	0.414	0.424	0.371	0.464	0.470	0.381	0.485	0.414	0.522	0.414	1.000	
EfficientnetB4	0.465	0.467	0.436	0.400	0.435	0.438	0.328	0.433	0.445	0.431	0.466	0.578	0.538	0.581	0.415	1.000
	ResNeXt101	Xception	InceptionResNetV2	SeResNeXt50	ResNeXt50	EfficientNetB0	SeResNeXt101	ResNet101	ResNet50	InceptionV3	EfficientnetB2	EfficientnetB5	EfficientnetB3	EfficinetnetB6	EfficientnetB1	EfficientnetB4

**Table 7 bioengineering-10-00069-t007:** True positive, false positive, false negative, and true negative values for 4-class classification in the models with the highest accuracy by pruning in each of the four types of ensemble learning. BLT: benign liver tumor, LCY: liver cyst, MLC: metastatic liver cancer, PLC: primary liver cancer, TP: true positive, FP: false positive, FN: false negative, TN: true negative, SV16: top16 was used for soft voting, ST9: top9 was used for stacking, WAV7: top7 was used for weighted average voting, WHV13: top13 was used for weighted hard voting.

		SV16	ST9	WAV7	WHV13
BLT	TP	194	188	196	194
	FP	52	53	57	49
	FN	56	62	54	56
	TN	698	697	693	701
LCY	TP	232	234	230	236
	FP	3	1	0	3
	FN	18	16	20	14
	TN	747	749	750	747
MLC	TP	185	184	184	184
	FP	96	99	93	97
	FN	65	66	66	66
	TN	654	651	657	653
PLC	TP	166	170	173	165
	FP	72	71	67	72
	FN	84	80	77	85
	TN	678	679	683	678

**Table 8 bioengineering-10-00069-t008:** Evaluation metrics for the models with the highest accuracy by pruning in each of the four ensemble learning methods. SV16: top16 was used for soft voting, ST9: top9 was used for stacking, WAV7: top7 was used for weighted average voting, WHV13: top13 was used for weighted hard voting.

Model	Precision *^,^**	Sensitivity *^,^**	Specificity *^,^**	F1 Score *	ROC Macro AUC	Accuracy **
SV16	0.783 (0.824, 0.733)	0.777(0.822, 0.724)	0.926 (0.941, 0.906)	0.779	0.94	0.777 (0.802, 0.750)
ST9	0.783 (0.822, 0.734)	0.776 (0.821, 0.723)	0.925 (0.940, 0.906)	0.778	0.924	0.776 (0.801, 0.749)
WAV7	0.790 (0.831, 0.749)	0.783 (0.832, 0.734)	0.928 (0.943, 0.912)	0.786	0.935	0.783 (0.809, 0.757)
WHV13	0.784 (0.829, 0.740)	0.779 (0.827, 0.731)	0.926 (0.943, 0.910)	0.781	–	0.779 (0.805, 0.753)

* macro-average ** (high and low 95% confidence interval).

**Table 9 bioengineering-10-00069-t009:** *p*-values of McNemar test between the single CNN model with the highest value of accuracy (ResNeXt101) and WAV from top2 to top16. WAV: weighted average voting.

*p*-Value	WAV2	WAV3	WAV4	WAV5	WAV6	WAV7	WAV8	WAV9	WAV10	WAV11	WAV12	WAV13	WAV14	WAV15	WAV16
ResNeXt101	0.192	0.037	0.014	0.003	0.001	0.001	0.001	0.001	0.001	0.004	0.001	0.003	0.001	0.001	0.001

**Table 10 bioengineering-10-00069-t010:** Metrics of 4 different liver masses for ResNeXt101, the model with the highest accuracy among 16 CNNs, and WAV7 (top7 was used for weighted average voting), the model with the highest accuracy among all models of ensemble learning. BLT: benign liver tumor, LCY: liver cyst, MLC: metastatic liver cancer, PLC: primary liver cancer.

ResNeXt101	Precision *	Sensitivity *	Specificity *	F1-Score	ROC AUC
BLT	0.745 (0.797, 0.685)	0.688 (0.742, 0.628)	0.921 (0.900, 0.939)	0.715	0.915
LCY	0.991 (1.000, 0.979)	0.904 (0.941, 0.867)	0.997 (1.000, 0.994)	0.946	0.994
MLC	0.567 (0.623, 0.510)	0.664 (0.723, 0.605)	0.831 (0.858, 0.804)	0.611	0.843
PLC	0.625 (0.685, 0.565)	0.620 (0.680, 0.560)	0.876 (0.900, 0.852)	0.622	0.854
WAV7					
BLT	0.775 (0.826, 0.723)	0.784 (0.835, 0.733)	0.924 (0.943, 0.905)	0.779	0.944
LCY	1.000 (1.000, 1.000)	0.920 (0.954, 0.886)	1.000 (1.000, 1.000)	0.958	0.999
MLC	0.664 (0.720, 0.609)	0.736 (0.791, 0.681)	0.876 (0.900, 0.852)	0.698	0.891
PLC	0.721 (0.778, 0.664)	0.692 (0.749, 0.635)	0.911 (0.931, 0.890)	0.706	0.903

*: high and low 95% confidence interval.

## Data Availability

The models are available upon request via this GitHub link: https://github.com/imedix2021/ame_liver_ensemble accessed on 20 November 2022. The original hepatic US dataset used in this study was selected from US images of hepatic mass lesions from the national project of “the construction of big database of US digital image: toward the diagnostic aid with artificial intelligence”, a part of the “ICT infrastructure establishment and implementation of artificial intelligence for clinical and medical research” fund (https://www.amed.go.jp/en/program/list/14/02/002.html accessed on 20 November 2022.), which is an effort to develop ICT infrastructure that utilizes the results of medical data analysis has been addressed in a fully integrated manner by the government of Japan. The contact information for this database is rinshoict@amed.go.jp.

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
