# Peer review of "Ensemble Learning of Multiple Models Using Deep Learning for Multiclass Classification of Ultrasound Images of Hepatic Masses"

_bioengineering, 2023, doi:10.3390/bioengineering10010069_

Round 1

Reviewer 1 Report

This manuscript reports a new study that investigate and compare 4 ensemble learning methods to combine 16 CNNs to classify 4 categories of liver lesions. Although this is a potentially interesting study, the manuscript is not well written. It lacks details in study background, hypothesis, dataset and methods. It is also difficult to understand how the reported study results are generated. It does not include correlation coefficients of 16 CNNs and confusion matrices of 4 category classification. Following are several of my specific comments.   

11. Introduction section is too short, and it should be expanded to provide more detailed literature review and background information of applying AI or deep learning models of ultrasound images. Are there any previous studies of applying deep learning models to detect and classify HCC of liver cancer? If not, then, this is the first such study in this field or topic. Please clarify this issue.  

22. The authors claimed that the previous ensemble learning was not optimal. Thus, the authors should clearly describe the scientific rationale of this new study to solve this problem in Introduction section. What is the hypothesis of this study?  

33. The description of dataset is unclear. The authors only reported total number of Ultrasound images. It does not report how many patients are involved in this dataset. If many images were acquired from the same patients, these images are highly correlated and redundant. Please report the number of patients in four categories. How many patients are involved in testing dataset?

44. Quality of Ultrasound images heavily varies greatly or heavily depends on the skill or preferences of the clinicians. Are any image preprocessing, standardization or normalization methods used and applied to the acquired clinical Ultrasound images?

55. What means “Noisy Student”? What image databases were used to build or train the original 16 CNNs? What is the scientific rationale to select these 16 CNNs in this study? 

66. In section 2.1. data set, the authors claimed to build an independent testing dataset with 1,000 images. However, in section 2.3. Training and Testing with 16 CNNs, the authors claimed that k-fold cross-validation (k=10) is used to validate performance of ML models. Which one is correct?

77. This study aims to classify 4 categories (classes) of liver lesions. ROC is used in classify two classes. Please clearly describe how to compute ROC curves in this study? 

88. Please provide scientific rationale of why choosing 4 ensemble learning methods in this study. How to optimally select and combing 16 CNNs?

99. The authors should compute and report correlations of prediction results or scores of 16 CNNs to demonstrate whether majority or some of CNNs are highly correlated or redundant.

110. The writing logic is confused and difficult to follow. For example, the figures, tables, and schemes in section 3.4 do not belong to Results section. They should be moved to Methods sections and provide more detailed explanation in Methods section.

111.  In Methods section, please describe how Heat Maps are generated. Are they generated using any CNN models? How these heat maps can be used to assist clinicians or can provide any quantitative analysis to help improve model predictive performance?

112. Please describe how the performance indices are computed in Table 2 using 1,000 testing images. How to compute standard deviations?

113. Please more clearly describe how 4 ensemble learning models are generated? Do they use all 16 CNNs and how to train these ensemble learning models?

114. The database includes 4 different categories of liver lesions. How AUC values, F1 scores and other performance indices are computed? How the TP, FP, TN and FN are defined. The authors should add and present confusion matrices of 4 categories of lesion classification to help readers understand the reported performance indices reported in Tables 2 – 4.

115. The authors should report patient or case-based predictive performance. Reporting image-based performance is not clinically useful. How many patients can correctly classified into 4 categories or classes of liver lesions?   

Author Response

Author's Reply to the Review Report (Reviewer 1)

  1. Introduction section is too short, and it should be expanded to provide more detailed literature review and background information of applying AI or deep learning models of ultrasound images. Are there any previous studies of applying deep learning models to detect and classify HCC of liver cancer? If not, then, this is the first such study in this field or topic. Please clarify this issue.

Answer.

I have added a detailed literature review and background on the application of AI and deep learning models in ultrasound imaging.

Manuscript, line 48.

“For this reason, a relatively large number of research papers have been published on the application of ultrasound-based AI to FLLs, distinguishing benign from malignant [17-22], HCC, metastatic liver tumors[23], or in addition to these, hemangiomas, FNH, non-neoplastic lesions, focal fatty infiltration, cyst, etc.[24-34] These papers mainly at-tempt image classification using CNN [17,18,25,26,34], other neural networks [24,28], support vector machine (SVM) [19,20,23], and radiomics [29-33].”

*FLLs: focal liver lesions.

  1. The authors claimed that the previous ensemble learning was not optimal. Thus, the authors should clearly describe the scientific rationale of this new study to solve this problem in Introduction section. What is the hypothesis of this study?

Answer.

Additional description of ensemble learning pruning.

Manuscript, line 59.

“Ensemble pruning methods, known as ensemble selection methods, aim at reducing the complexity of ensemble models. One of the typical pruning methods is the ranking method. The ranking method rank the individual members based on a predetermined criterion and select the top ranked base models according to a threshold [49]. There are studies that performed ensemble learning by selecting the top 3 models out of 9 trained CNN models [35], and a study using 15 trained CNNs and ensemble learning from all of them [37]. Both of these two papers use the ranking method, but neither provides a ra-tionale as to why the optimal rank was selected, and there is no discussion of ensemble pruning. In a study of deep ensemble learning for COVID-19 detection, the results of all seven learning models confirmed that the performance of the top three ensembles was superior to that of the top five and top seven ensembles [47]. “

The hypothesis was also added as follows. Manuscript, line 70.

“The purposes of this study are as follows. 1) To compare the usefulness of several ensemble learning methods using multiple CNN learning models for multi-class image classification of liver masses in US images. 2) To confirm the hypothesis that top3 is the best performing ensemble learning method for multi-class image classification of US FFLs in terms of ranking method, which is one of the ensemble pruning methods.”

  1. The description of dataset is unclear. The authors only reported total number of Ultrasound images. It does not report how many patients are involved in this dataset. If many images were acquired from the same patients, these images are highly correlated and redundant. Please report the number of patients in four categories. How many patients are involved in testing dataset?

Answer.

The number of patients was added.

Manuscript, line 82.

“In this study,101,242 B-mode images of FFLs from 30,873 cases were collected from April 1, 2018 to July 25, 2021 in 11 core medical centers through a research project funded by the Japan Agency for Medical Research and Development (AMED), entitled "National Database Construction of Digital Ultrasound Images and Artificial Intelligence-Assisted Ultrasound Diagnostic System Development" [16]. Of these images, this study included 26,440 ultrasound B-mode, grayscale still images of 6,056 cases of FLL that were selected. The images used in this project were classified into four categories: benign liver tumor (BLT) 6,320 images from 1564 cases, liver cyst (LCY) 2,560 images from 1018 cases, metastatic liver cancer (MLC) 9,720 images from 1469 cases, primary liver cancer (PLC) 7,840 images from 2005 cases.”

Once all eligible images (26,440) had been randomly sorted, 1000 of them were selected again at random as test images. This made it difficult to track which patient the 1000 images belonged to.

  1. Quality of Ultrasound images heavily varies greatly or heavily depends on the skill or preferences of the clinicians. Are any image preprocessing, standardization or normalization methods used and applied to the acquired clinical Ultrasound images?

Answer.

All images were converted to 256x256 pixels and then standardized in the python program.

  1. What means “Noisy Student”? What image databases were used to build or train the original 16 CNNs? What is the scientific rationale to select these 16 CNNs in this study?

Answer1.

For EfficientNetB0-B6, instead of ImageNet, we used ImageNet trained with the Noisy studemt method as a trained model.

  1. the teacher model (in this case, ImageNet) is trained with labeled data only. 2.

Pseudo-labels are attached to unlabeled data in the teacher model. 3.

Prepare a student model that is the same as or larger than the teacher model. 4.

  1. train the student model with labeled + pseudo-labeled data and noise.

The noise is 1: Rand Augmentation→noise to the input image, 2: Dropout→noise to the model, 3: Stochastic depth→noise to the model, and 4: Stochastic depth→noise to the model.

3: Stochastic depth→noise to the model.

It has been reported that using this Noisy student method improves performance in EfficientNet compared to learning with conventional ImageNet (manuscript text, line 750, reference paper number 51).

In this study, we used this ImageNet-trained model by Noisy studemt for EfficienNet.

Manuscript text, line 114 .

“EfficientNetB0-B6 [50] were pre-trained on ImageNet with Noisy Student, a kind of semi-supervised learning [51]…..”

Answer2.

In this study, as a preliminary experiment, we conducted experiments using 22 training models, almost all of which were available as Keras applications at the time the study was launched in 2021. As a result, VGG16, VGG19, MobileNetV2, DenseNet, NasNetMobile, and NasNetLarge were excluded due to their poor performance.

  1. In section 2.1. data set, the authors claimed to build an independent testing dataset with 1,000 images. However, in section 2.3. Training and Testing with 16 CNNs, the authors claimed that k-fold cross-validation (k=10) is used to validate performance of ML models. Which one is correct?

Answer.

It is illustrated in detail in Figure 1, which has been added.

The k-fold cross-validation was used for validation during training; testing with 1,000 copies of the Test Set was performed independently.

  1. This study aims to classify 4 categories (classes) of liver lesions. ROC is used in classify two classes. Please clearly describe how to compute ROC curves in this study?

Answer.

Manuscript, line 189.

‘The receiver operating characteristic (ROC) Curve and the ROC area under the curve (AUC) score are important tools to evaluate binary classification models. But this concept is not immediately applicable for multiclass classifiers. OvR stands for “One vs Rest”, and as the name suggests is one method to evaluate multiclass models by comparing each class against all the others at the same time. By using this OvR, we reduce the multiclass classification output into a binary classification one, and it is possible to use all the known binary classification metrics to evaluate the multiclass classification[53,54].’

Reference 54

Evaluation of Multi-Class Classification by ROC Curves and ROC AUCs|by Vinícius Trevisan Toward Data Science

https://towardsdatascience.com/multiclass-classification-evaluation-with-roc-curves-and-roc-auc-294fd4617e3a

I cited OvR - One vs Rest for a clear explanation.

  1. Please provide scientific rationale of why choosing 4 ensemble learning methods in this study. How to optimally select and combing 16 CNNs?

Answer.

Manuscript, line 214.

“ince most of the previous papers [35-48] on ensemble learning with CNNs for medical images used either simple soft voting (SV), weighted average voting (WAV), weighted hard voting (WHV), or stacking(ST) (Figure2), we used four types of ensemble learning in this study: SV, WAV, WHV, and ST. Ensemble learning includes voting, bagging, boosting, and ST [56-58]. Voting includes SV and simple hard voting (majority voting). Hard voting is a simple majority voting image classification in which each classifier has one vote. In the case of hard voting, in order to obtain the optimal number of en-semble pruning based on the ranking method described in the next section, using an even number of classifiers may result in a tie number of votes for the four classes, and the same number of votes may make the class determination for image classification impossible. To avoid this situation and improve the performance of ensemble learning, there is a technique called WHV, also known as Weighted Majority Voting, which as-signs different weights to each classifier [59]. WHV is a method that assigns different weights to each classifier [59]. SV also has a method called WAV, which aims to im-prove performance by optimizing each classifier with different weights [48]. There are also methods for weight optimization, such as grid search, random search, and Bayes-ian optimization [60,61]. In this study, weighting optimization was performed using Optuna, Optuna version 2.10.0 (https://www.preferred.jp/ja/projects/optuna/), an open-source software framework for automating the optimization of hyperparameters, which is a type of Bayesian optimization for both WHV and WAV. There are also stud-ies that use extreme gradient boosting (XGBoost), a representative method of gradient boosting, as a meta-learner for ST [62,63]. In this study, we employed the ST method using light gradient boosting machine (LightGBM) version 3.2.1 (https:// lightgbm.readthedocs.io/) as the meta-learner, which is a newer method of gradient boosting than XGBoost and is expected to have higher performance [63,64].”

  1. The authors should compute and report correlations of prediction results or scores of 16 CNNs to demonstrate whether majority or some of CNNs are highly correlated or redundant.

Answer.

Table 5 lists the p-values, which are the results of McNemar tests on the difference in performance between ResNeXt, which ranked first, and the other 15 models.

Original manuscript, line 350.

“Comparing the test results of top1, ResNeXt101, with those of the other 15 CNN models, there was no significant difference between top2 and top9, and a clear significant dif-ference between top1 and top10 to top16 (p<0.05) (Table5).”

  1. The writing logic is confused and difficult to follow. For example, the figures, tables, and schemes in section 3.4 do not belong to Results section. They should be moved to Methods sections and provide more detailed explanation in Methods section.

Answer.

All figures and tables have been reviewed and rearranged.

  1. In Methods section, please describe how Heat Maps are generated. Are they generated using any CNN models? How these heat maps can be used to assist clinicians or can provide any quantitative analysis to help improve model predictive performance?

Answer.

Manuscript, line 482.

. Four classes of heatmaps were created using ResNeXt101 and GradCAM, the model with the highest accuracy among the 16 CNNs, and representative examples are shown in Figure 9.

  1. Please describe how the performance indices are computed in Table 2 using 1,000 testing images. How to compute standard deviations?

Answer.

Heatmap is presented to confirm that the CNN's area of interest for the lesions in this study is correct. Its clinical usefulness and contribution to improved predictive performance is considered outside the scope of this study. Therefore, we believe that these questions need to be addressed in a separate study.

  1. Please more clearly describe how 4 ensemble learning models are generated? Do they use all 16 CNNs and how to train these ensemble learning models?

Answer.

Answer.

Figure 2 was used and the author explained the parameters and other details as much as possible in the text of lines 214-245 of the manuscript.

  1. The database includes 4 different categories of liver lesions. How AUC values, F1 scores and other performance indices are computed? How the TP, FP, TN and FN are defined. The authors should add and present confusion matrices of 4 categories of lesion classification to help readers understand the reported performance indices reported in Tables 2 – 4.

Answer1. How to define AUC values, F1 scores and other performance indices.

The manuscript, lines 167-188, explained.

Answer2. How to define the TP, FP, TN and FN.

The manuscript, lines 131-165, explained.

Answer3. Confusion matrices of 4 categories of lesion classification.

Figure4, Figure7 were added.

  1. The authors should report patient or case-based predictive performance. Reporting image-based performance is not clinically useful. How many patients can correctly classified into 4 categories or classes of liver lesions?

Answer.

The author is a radiologist (diagnostic imaging physician). The job of radiologists is to read images and write reading reports. Scientific activities are also based on this philosophy.

Ultrasound diagnosis of liver masses is clinically important. For this reason, many studies by AI are considered to be published (Manuscript References [16-34]).

Reviewer 2 Report

This manuscript proposed a novel deep learning-based method for multi-class classification of ultrasonic images of hepatic masses, where ensemble learning of multiple convolutional neural networks (CNN) was developed for the task of interest. The performance of the proposed method has been evaluated using a large number of ultrasonic images, with satisfactory results via the comparison with individual CNN model. Overall, the topic of this research is interesting, and the manuscript was well organised and written. The detailed comments are provided as follows.

1.       The contribution and innovation of the manuscript should be clarified clearly in abstract and introduction.

2.       Broaden and update literature review on CNN or ensemble CNNs and their practical applications, such as image processing, data processing, etc. E.g. Torsional capacity evaluation of RC beams using an improved bird swarm algorithm optimised 2D convolutional neural network; Vision-based concrete crack detection using a hybrid framework considering noise effect.

3.       The performance of CNN is mainly dependent on the setting of hyperparameters. How did the authors set network parameters in this research to achieve the optimal classification performance?

4.       In this study, the authors used 16 CNNs with different achitectures to develop the classification model, including Xception, Inception, ResNet, etc. However, these networks all belong to pre-trained CNNs, which were developed for other tasks. Accordingly, I think transfer learning has been used in this study to transfer these pre-trained networks to networks that can be used in the classification of ultrasonic images of hepatic masses. If so, more contentsabout transfer learning should be added for the clarification.

5.       In general, confusion matrix is always used for demonstrating performance of machine learning models for multi-classification. Please add the confusion matrix, if possible.

6.       How about the robustness of the proposed method against noise effect, which may be contained in the image.

7.       More future research should be included in conclusion part.

Author Response

  1. The contribution and innovation of the manuscript should be clarified clearly in abstract and introduction.

Answer. Added a description of ensemble pruning to the purpose section of the abstract and added a substantial Introduction.

  1. Broaden and update literature review on CNN or ensemble CNNs and their practical applications, such as image processing, data processing, etc. E.g. Torsional capacity evaluation of RC beams using an improved bird swarm algorithm optimised 2D convolutional neural network; Vision-based concrete crack detection using a hybrid framework considering noise effect.

Answer. The author have greatly expanded our literature review on CNNs and ensemble CNNs and their practical applications in image and data processing.

  1. The performance of CNN is mainly dependent on the setting of hyperparameters. How did the authors set network parameters in this research to achieve the optimal classification performance?

Answer.

Manuscript, line 111

Description of parameters for 16 CNNs.

“The resolution of all input images was set to 256x256 in order to compare them and ensure reproducibility of training results. The batch sizes were all set to 32, the number of epochs to 50, and the random number seed was fixed to an arbitrary seed value. Width shift, height shift and horizontal flip were used as data augmentation. The float-ing-point number for width shift and height shift was set to 0.1. K-partition cross-validation (k=10) was used as a method to validate the predictive performance of the machine learning models (Figure1). The model fitting callback was the early stop-ping function of Keras, which stops training before overtraining occurs, and training was stopped if there was no improvement during 10 epochs of val_loss values. We also reduced the learning rate by 0.1 if there was no improvement for 3 epochs using Re-duceLROnPlateau in Keras.”

Since Tensorflow and Keras are used in this study, parameters other than those listed are already included in the respective trained models.

  1. In this study, the authors used 16 CNNs with different achitectures to develop the classification model, including Xception, Inception, ResNet, etc. However, these networks all belong to pre-trained CNNs, which were developed for other tasks. Accordingly, I think transfer learning has been used in this study to transfer these pre-trained networks to networks that can be used in the classification of ultrasonic images of hepatic masses. If so, more contentsabout transfer learning should be added for the clarification.

Answer.

As shown in line 111 of the text, we either replace all the coupled layers or fix some of the convolution layers. 16 different individual CNNs will determine which layers to fix, and we have published the source on GitHub.

https://github.com/imedix2021/ame_liver_ensemble

  1. In general, confusion matrix is always used for demonstrating performance of machine learning models for multi-classification. Please add the confusion matrix, if possible.

Answer. Figure 4 and Figure 7 were added and the confusion matrix was placed in the manuscript.

  1. How about the robustness of the proposed method against noise effect, which may be contained in the image.

Answer.  Originally, ultrasound images are speckle images, with many artifacts and noise. Naturally, it is thought that there is robustness due to those features, but since deep learning is considered to have an advantage in learning such diverse images, we expect to see results that include those features in this case. However, since how robustness affects AI diagnosis is away from the main objective of the study in this case, we believe that a separate comparative study should be conducted depending on the presence of noise.

  1. More future research should be included in conclusion part.

Answer.

Manuscript, line 581. The following text was added to the conclusion section.

“It was thought that future research should explore ensemble pruning for medical im-ages such as X-ray, CT and MRI, other than US images, especially to find out how many classifier members are useful in ensemble learning by ranking methods.”

Round 2

Reviewer 1 Report

The authors made many revisions and expansion in the revised manuscript. However, the revised manuscript still lacks detailed information of the study methods and clear description or discussion of study results. Responses to the following comments made in my previous review are not acceptable. May issues remain.

1.     In response to previous Comment 2, the authors wrote “to confirm the hypothesis that top3 is the best performing ensemble learning method for multi-class image classification of US FFLs in terms of ranking method, which is one of the ensemble pruning methods.” What means “top3”? The reported study results do not support/confirm above hypothesis. Table 4 shows that top 3 models are ResNeXt101, Xception and InceptionResNetV2. However, Data reported in Table 7 shows 4 ensemble learning methods that all use more than top 3 models. 

2.     In response to previous Comment 3, the authors wrote “the number of patients was added.” However, the revised manuscript does not clearly describe how many patients are included. For example, it is still unknown 1,000 testing images were acquired from how many patients? Were these 1,000 images acquired from 1,000 different patients? If not, what will be the patient-based classification performance or how to address potential bias issues of the images acquired from the same lesions of the same patients?

3.     In response to previous Comment 4, the authors wrote “All images were converted to 256x256 pixels and then standardized in the python program.” The authors did not answer my question. Python program is just a computer software platform. It is not an image standardization method.

4.     In response to previous Comment 5, the authors still do not clearly describe or discuss what means “Noisy Student” in the revised manuscript. The response such as “we conducted experiments using 22 training models, almost all of which were available as Keras applications at the time the study was launched in 2021. As a result, VGG16, VGG19, MobileNetV2, DenseNet, NasNetMobile, and NasNetLarge were excluded due to their poor performance” is irrelevant to my comment or question.

5.     In response to previous Comment 7, the authors added too many unnecessary expansions in the revised manuscript. The authors only need to add one or two sentences to describe ROC curves are generated based on “One vs. Rest” definition. Please delete Table 1 and related lengthy descriptions.

6.     In response to previous Comment 9, the authors claimed that they have computed p-values using ResNeXt101 as a baseline. However, correlation coefficients are different from p-values. For example, performances of top 3 models (ResNeXt101, Xception and InceptionResNetV2) are not significantly different. However, it is unknow whether the prediction scores generated by these top 3 models are highly correlated or not. If they are highly correlated, ensemble method using these top 3 models does not help to increase prediction performance.

7.     In response to previous Comment 11, the authors still do not provide any study results or data to demonstrate that “heat maps” can help more accurately diagnose hepatic masses. The heat maps may be useful in observer performance study, but they are irrelevant to this study. Please delete this section and figure 9.

8.     In response to previous Comment 12, the authors wrote “Heatmap is presented to confirm that the CNN's area of interest for the lesions in this study is correct. Its clinical usefulness and contribution to improved predictive performance is considered outside the scope of this study. Therefore, we believe that these questions need to be addressed in a separate study.” The answer is irrelevant to my comment or question.

9.     In response to previous Comment 13, the authors did not provide any response. Just skip this comment.

10.  In response to previous Comment 15, the authors wrote “The author is a radiologist (diagnostic imaging physician). The job of radiologists is to read images and write reading reports. Scientific activities are also based on this philosophy.” This may be true if the authors can clearly describe images used in this study. If multiple images are acquired from the same mass (such as at different orientations) and some images are classified as positive while others are classified as negative, how such an AI tool can help radiologists? 

Additional Comments – In the revised manuscript, the authors added several new Tables or Figures in the Results section, which makes reading this paper more difficulty or confused. For example,

11.  The authors add a new figure (Figure 6) in Results section to show 4 curves. However, the authors did not describe what method is used to find the top N models in 4 ensemble learning methods. Did the authors use “an exhaustive searching” method or a machine learning method (i.e., a genetic algorithm)? What training, validation and testing datasets are used to define the optimal top models? For example, in WAV7, what are the 7 models? Are they top 7 models listed in Table 4?   

12.  The authors added several redundant tables and/or figures. For example, Table 3 and Figure 4 are redundant and report the same information. Please delete one of them.

13.  The curves in Figure 5 are also redundant with the data reported in Tables 4 and 5. The curves do not provide new information. Additionally, the image quality of the figure is also low. Please delete Figure 5.

14.  It is unclear or confused what is the purpose to add Tables 8 and 9. Why need to report so many p-values? For example, in the first (top) section of Table 9, all p-values are not statistically significantly different (p > 0.05). Does it mean that all ensemble learning using 2 to 16 models are the same?     

Author Response

  1. In response to previous Comment 2, the authors wrote “to confirm the hypothesis that top3 is the best performing ensemble learning method for multi-class image classification of US FFLs in terms of ranking method, which is one of the ensemble pruning methods.” What means “top3”? The reported study results do not support/confirm above hypothesis. Table 4 shows that top 3 models are ResNeXt101, Xception and InceptionResNetV2. However, Data reported in Table 7 shows 4 ensemble learning methods that all use more than top 3 models.

Response1:

The second hypothesis was modified as follows.

Linn74-  2) To investigate the usefulness of one of the ensemble pruning methods, the ranking method, for multiclass image classification of FFLs in US images.

  1. In response to previous Comment 3, the authors wrote “the number of patients was added.” However, the revised manuscript does not clearly describe how many patients are included. For example, it is still unknown 1,000 testing images were acquired from how many patients? Were these 1,000 images acquired from 1,000 different patients? If not, what will be the patient-based classification performance or how to address potential bias issues of the images acquired from the same lesions of the same patients?

Response2:  The following statement was added to identify the patient to whom the 1000 test images belong.

And For each class, 250 images were checked for the absence of similar images to address the bias issues.

Line99-  The 250 test images in each of the four classes were collected from 227 BLT cases, 200 LCY cases, 210 MLC cases, and 191 PLC cases. In order to check for similarity in test images of lesions collected from the same case, the test images for each class were hashed and the Hamming distance was measured to ensure that there were no similar images. Similarity was judged as not similar if the bit agreement between the hash values of the two images was less than 80% [50].

  1. In response to previous Comment 4, the authors wrote “All images were converted to 256x256 pixels and then standardized in the python program.” The authors did not answer my question. Python program is just a computer software platform. It is not an image standardization method.

Response3: The following text was added.

Line105- Pixel information in US grayscale images is provided by numbers ranging from 0 to 255, with 0 being white and the numbers becoming black as they approach 255. All US image data used were standardized by arraying, converting the data type to a 64-bit floating-point number, and then dividing by 255, with the pixel information being a number in the range 0 to 1.

  1. In response to previous Comment 5, the authors still do not clearly describe or discuss what means “Noisy Student” in the revised manuscript. The response such as “we conducted experiments using 22 training models, almost all of which were available as Keras applications at the time the study was launched in 2021. As a result, VGG16, VGG19, MobileNetV2, DenseNet, NasNetMobile, and NasNetLarge were excluded due to their poor performance” is irrelevant to my comment or question.

Response4: The following sentences were added to the Methods and Discussion, respectively.

Line124- Noisy Student Training extends the idea of self-training and distillation with the use of equal-or-larger student models and noise added to the student during learning. ImageNet first trains an EfficientNet model on the labeled images, which are used as a teacher to generate pseudo-labels for the 300 million unlabeled images. Next, a larger EfficientNet is trained as a student model for the combination of labeled and pseu-do-labeled images. This process is iterated by returning the student as the teacher. During student learning, inject noise into the student, such as dropouts, probabilistic depth, and data augmentation with RandAugment, to ensure that the student general-izes better than the teacher [52].

Line475-   EfficientNet is newer than the resnet and inception-based CNN models, and EfficientNet is generally expected to perform better in ImageNet with Noisy Student training than ImageNet without Noisy Student [52]. For this reason, we employed ImageNet with noisy student training for pre-training of EfficientNet.

  1. In response to previous Comment 7, the authors added too many unnecessary expansions in the revised manuscript. The authors only need to add one or two sentences to describe ROC curves are generated based on “One vs. Rest” definition. Please delete Table 1 and related lengthy descriptions.

Response5: Table 1 was deleted and shortened to the following text.

Line203-   The receiver operating characteristic (ROC) curve and ROC area under the curve (AUC) score are not immediately applicable to a multiclass classifier. One vs. Rest, where each class is compared to the other classes simultaneously, allows drawing ROC curves for multiclass classifiers. 16 CNNs with 4 classes had ROC curves AUCs computed. Macro averages of the AUCs were then also computed for each model [56,57]. Finally, for each of the 16 CNN models, the macro-averaged accuracy values of the test results were ranked and sorted from highest to lowest.

  1. In response to previous Comment 9, the authors claimed that they have computed p-values using ResNeXt101 as a baseline. However, correlation coefficients are different from p-values. For example, performances of top 3 models (ResNeXt101, Xception and InceptionResNetV2) are not significantly different. However, it is unknow whether the prediction scores generated by these top 3 models are highly correlated or not. If they are highly correlated, ensemble method using these top 3 models does not help to increase prediction performance.

Response6: The following statements regarding the correlation coefficients of the test results of 16 different CNNs were added to the Methods, Results, and Discussion.

Line210- 2.5 Correlation matrix for each of the 16 CNN test results

   A quantitative measure of the ensemble's effectiveness is to look at the Pearson's correlation coefficient (CORR) between classifiers. The CORRs of the estimated labels of 16 CNN classifiers are calculated.

Line358-   The CORRs among the estimated labels of the 16 CNN classifiers totaled 120. Of these, 8 were in the 0.3 range and 112 were 0.4 or higher. All 120 CORRs were below 0.95 (Table 6).

Line487-   A quantitative measure of the effectiveness of ensemble is to look at the Pearson's CORR between classifiers, and it is said that ensemble learning is expected to be effective when Pearson's CORR is 0.95 or less [70]. The 16 CNNs used in this study all had CORRs of 0.95 or less, suggesting that ensemble learning can help improve prediction performance.

  1. In response to previous Comment 11, the authors still do not provide any study results or data to demonstrate that “heat maps” can help more accurately diagnose hepatic masses. The heat maps may be useful in observer performance study, but they are irrelevant to this study. Please delete this section and figure 9.

Response7: Deleted pertinent sections and Figure 9.

  1. In response to previous Comment 12, the authors wrote “Heatmap is presented to confirm that the CNN's area of interest for the lesions in this study is correct. Its clinical usefulness and contribution to improved predictive performance is considered outside the scope of this study. Therefore, we believe that these questions need to be addressed in a separate study.” The answer is irrelevant to my comment or question.

Response8: Calculated 95% confidence interval instead of standard deviation. Table 6 omitted that statement, so we included it in the notes and added the following statement to the method.

Line201-   The 95% confidence intervals for precision, sensitivity, specificity, and accuracy were all calculated using the Newcombe method [55].

  1. In response to previous Comment 13, the authors did not provide any response. Just skip this comment.

Response9:

Methods

Line223-.

2.6.1.1 Voting Ensemble including WHV, SV, and WAV             

The text in the above paragraph has been completely revised and the formulas for WHV and WAV have been added.

Also added revised Figures 3 and 4.

  1. In response to previous Comment 15, the authors wrote “The author is a radiologist (diagnostic imaging physician). The job of radiologists is to read images and write reading reports. Scientific activities are also based on this philosophy.” This may be true if the authors can clearly describe images used in this study. If multiple images are acquired from the same mass (such as at different orientations) and some images are classified as positive while others are classified as negative, how such an AI tool can help radiologists?

Response10: The following text was added at the end of the discussion.

Line512-   Although very rare, cases of PLC associated with BLT have been reported, and US images have been included in these case reports [71,72]. It is believed that the general DL, including this study, cannot correctly diagnose such rare cases, which is a limitation of this study. Further studies are needed for computer assisted diagnosis of such rare diseases. However, if the DL model predicts BLT on one scan and PLC on another scan for the same lesion, it may provide a hint for radiologists to consider the possibility of rare cases for the paradoxical prediction of DL.

Additional Comments – In the revised manuscript, the authors added several new Tables or Figures in the Results section, which makes reading this paper more difficulty or confused. For example,

  1. The authors add a new figure (Figure 6) in Results section to show 4 curves. However, the authors did not describe what method is used to find the top N models in 4 ensemble learning methods. Did the authors use “an exhaustive searching” method or a machine learning method (i.e., a genetic algorithm)? What training, validation and testing datasets are used to define the optimal top models? For example, in WAV7, what are the 7 models? Are they top 7 models listed in Table 4?

Response11: This method is the Ranking method and is used in the sentence Line60-.

Ensemble pruning methods, known as ensemble selection methods, aim at reducing the complexity of ensemble models. One of the typical pruning methods is the ranking method. The ranking method rank the individual members based on a predetermined criterion and select the top ranked base models according to a threshold [49].

Since this explanation alone is not clear, we have added an explanation below.

Added explanation using ST9 as an example.

Line377-   For ST, top9 (ST9) had the highest accuracy at 0,776. Referring to Table 5, this ST9 refers to ST using a total of nine different classifiers: ResNeXt101, Xception, InceptionResNetV2, SeResNeXt50, ResNeXt50, EfficientNetB0, SeResNeXt101, ResNet101, and ResNet50, starting from top1.

  1. The authors added several redundant tables and/or figures. For example, Table 3 and Figure 4 are redundant and report the same information. Please delete one of them.

Response12: Figure 4 in the original manuscript has been deleted.

  1. The curves in Figure 5 are also redundant with the data reported in Tables 4 and 5. The curves do not provide new information. Additionally, the image quality of the figure is also low. Please delete Figure 5.

Response13: Figure 5 and 8 in the original manuscript has been deleted.

  1. It is unclear or confused what is the purpose to add Tables 8 and 9. Why need to report so many p-values? For example, in the first (top) section of Table 9, all p-values are not statistically significantly different (p > 0.05). Does it mean that all ensemble learning using 2 to 16 models are the same?

Response14: Table 8 keeps only WAV, which performed best, and removes rows SV, ST, WAVST, and WHV.

Table 9 in the original manuscript was deleted.

Reviewer 2 Report

All the comments have been well addressed.

Author Response

Spell check was done over and over.

Round 3

Reviewer 1 Report

The responses are in general satisfactory and the quality of manuscript has significantly improved. I think that this will be an interesting study of applying DL models to Ultrasound images. Best wishes for the continuous success of the authors' research work.